# Routine, Cost-Effective SARS-CoV-2 Surveillance Testing Using Pooled Saliva Limits Viral Spread on a Residential College Campus

[ID] Nicole A. Vander Schaaf,[a] Anthony J. Fund,[a] Brianna V. Munnich,[a] Alexi L. Zastrow,[a]* Erin E. Fund,[a]§ Tanner L. Senti,[a] Abigail F. Lynn,[a] Jonathon J. Kane,[a] Jennifer L. Love,[a] Gregory J. Long,[a] Nicholas J. Troendle,[a] [ID] Daniel R. Sharda[a]

[a]Department of Biological Sciences, Olivet Nazarene University, Bourbonnais, Illinois, USA

**ABSTRACT** Routine testing for SARS-CoV-2 is rare for institutes of higher education due to prohibitive costs and supply chain delays. During spring 2021, we routinely tested all residential students 1 to 2 times per week using pooled, RNA-extraction-free, reverse transcription quantitative PCR (RT-qPCR) testing of saliva at a cost of $0.43/sample with same-day results. The limit of detection was 500 copies/ml on individual samples, and analysis indicates 1,000 and 2,500 copies/ml in pools of 5 and 10, respectively, which is orders of magnitude more sensitive than rapid antigen tests. Importantly, saliva testing flagged 83% of semester positives (43,884 tests administered) and was 95.6% concordant with nasopharyngeal diagnostic results (69.0% concordant on the first test when the nucleocapsid gene (N1) cycle threshold ($C_T$) value was >30). Moreover, testing reduced weekly cases by 59.9% in the spring despite far looser restrictions, allowing for more normalcy while eliminating outbreaks. We also coupled our testing with a survey to clarify symptoms and transmissibility among college-age students. While only 8.5% remained asymptomatic throughout, symptoms were disparate and often cold-like (e.g., only 37.3% developed a fever), highlighting the difficulty with relying on symptom monitoring among this demographic. Based on reported symptom progression, we estimate that we removed 348 days of infectious individuals by routine testing. Interestingly, viral load ($C_T$ value) at the time of testing did not affect transmissibility ($R^2 = 0.0085$), though those experiencing noticeable symptoms at the time of testing were more likely to spread the virus to close contacts (31.6% versus 14.3%). Together, our findings support routine testing for reducing the spread of SARS-CoV-2. Implementation of cost- and resource-efficient approaches should receive strong consideration in communities that lack herd immunity.

**IMPORTANCE** This study highlights the utility of routine testing for SARS-CoV-2 using pooled saliva while maintaining high sensitivity of detection (under 2,500 copies/ml) and rapid turnaround of high volume (up to 930 samples in 8 h by two technicians and one quantitative PCR [qPCR] machine). This pooled approach allowed us to test all residential students 1 to 2 times per week on our college campus during the spring of 2021 and flagged 83% of our semester positives. Most students were asymptomatic or presented with symptoms mirroring common colds at the time of testing, allowing for removal of infectious individuals before they otherwise would have sought testing. To our knowledge, the total per-sample consumable cost of $0.43 is the lowest to date. With many communities still lagging in vaccination rates, routine testing that is cost-efficient highlights the capacity of the laboratory's role in controlling the spread of SARS-CoV-2.

**KEYWORDS** COVID-19, SARS-CoV-2, surveillance, pooled, saliva, RT-qPCR, residential living, surveillance studies

The COVID-19 pandemic has greatly challenged the ability of schools, universities, and workplaces to conduct face-to-face activities safely. Although masking, distancing, and now vaccinations have helped reduce the number of COVID-19 cases, the spread of

**Ad Hoc Peer Reviewer** [ID] Luciana Costa, Universidade Federal do Rio de Janeiro; Akshat Mullerpatan

Address correspondence to Nicole A. Vander Schaaf, navanderschaaf@olivet.edu, or Daniel R. Sharda, drsharda@olivet.edu.

*Present address: Alexi L. Zastrow, Mayo Clinic Graduate School of Biomedical Sciences, Rochester, Minnesota, USA.

§Present address: Erin E. Fund, University of Illinois College of Medicine, Peoria, Illinois, USA.

Pooled saliva testing makes weekly routine COVID screening feasible for entire residential college campus.

SARS-CoV-2 will continue to be a concern throughout the coming months due to a substantial unvaccinated population locally and worldwide, breakthrough infections, new variants, and waning immunity. Vaccination rates are particularly lagging among young individuals such as college students, with only 49.8% of 18- to 24-year-olds being fully vaccinated as of September 14, 2021 (1).

Prior to the widespread availability of vaccinations in the United States, a few universities demonstrated that routine COVID-19 testing of students, in addition to the usual mitigation strategies, was effective in managing viral spread on residential campuses. For instance, Duke University implemented mandatory, twice-weekly testing of residential undergraduates during the fall 2020 semester, which kept COVID-19 case counts relatively low with an average weekly per capita infection incidence of 0.08% (2). Clemson University began their fall 2020 semester with a surveillance-based informative testing program, estimating a potential 154% increase in COVID-19 cases on campus had only voluntary testing been conducted (3). Modeling studies also support the effectiveness of regular COVID-19 testing in a university setting for controlling viral spread, with one study estimating that 96% of infections could be prevented by routine testing (4, 5). It is no surprise, therefore, that organizations such as the Centers for Disease Control and Prevention (CDC) and the American College Health Association (ACHA) have encouraged routine testing at institutes of higher education (6, 7).

A key strength of regular testing is that it identifies asymptomatic or presymptomatic individuals and isolates them before they spread the virus. Although the exact prevalence of asymptomatic COVID-19 cases among college-aged individuals is not well established, meta-analyses estimate that 17 to 20% of SARS-CoV-2-infected individuals remain asymptomatic throughout their infection (8, 9), making symptom tracking ineffective for identifying these individuals. The remaining ~80% will likely still propagate viral spread, especially if their symptoms remain mild or common cold-like and thus do not prompt COVID-19 testing. Viral spread among college students is not limited to the residential campus, but rather, extends to the off-campus community through jobs, internships, or visiting family members. Thus, the benefits of routine testing extend beyond the campus walls.

One hurdle to large-scale, routine testing of a population is the cost in time and money. Although antigen tests offer a less-expensive alternative and faster results compared to reverse transcription quantitative PCR (RT-qPCR) testing, they suffer from significant loss of sensitivity. For instance, one study found that Abbott BinaxNOW Rapid Antigen tests have a sensitivity of 64.2% for specimens from symptomatic persons and 35.8% for those from asymptomatic persons compared to RT-qPCR testing (10). Thus, RT-qPCR testing is ideal for maximal sensitivity of SARS-CoV-2 detection. However, routine testing of a population of individuals by RT-qPCR is expensive to the point of being cost-prohibitive for some universities; testing strategies that reduce the cost while maintaining high sensitivity are needed.

Here, we report the outcomes of routine, RT-qPCR, SARS-CoV-2 surveillance testing on the campus of Olivet Nazarene University (ONU), a small liberal arts university in the midwestern United States, for limiting virus spread on a residential campus during the spring 2021 semester. During the fall 2020 semester, viral containment was exclusively through isolation of self-reporting, symptomatic individuals. This resulted in outbreaks on campus, including on athletic teams and other extracurricular groups, and heightened social restrictions (closing all common areas, prohibiting visits to any dorm room other than one's own, etc.). We hypothesized that regular surveillance testing of all residential students would substantially reduce the spread of SARS-CoV-2 on campus during the spring semester, protect the health of the campus and local community, and also enable a loosening of the stifling social restrictions. We then used a survey to collect data from SARS-CoV-2 saliva-positive individuals to investigate the relationship between viral load, symptoms, and contagiousness among college-aged students, which we report herein.

Although nasopharyngeal samples have most commonly been used to date for SARS-CoV-2 detection, we chose to use saliva, a safe and effective alternative to the invasive nasopharyngeal swabs. The sensitivity, specificity, and viral load of SARS-CoV-2 detection by RT-qPCR from saliva are comparable to, and in some reports more sensitive than, those from paired nasopharyngeal swabs from infected individuals (11–13), making it an attractive

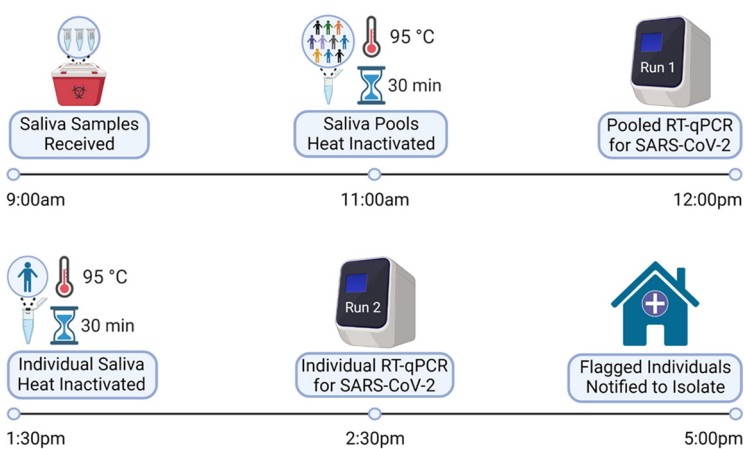

**FIG 1** Daily workflow for SARS-CoV-2 surveillance testing. Saliva samples were deposited by students into drop boxes around campus as early as 2 p.m. the day before their assigned test and collected at 9 a.m. each testing day. After the barcodes of each sample were scanned, saliva was pooled in groups of 5 or 10 individuals and heat-inactivated. Processed saliva was then mixed with assay reagents for detection of SARS-CoV-2 on a QuantStudio 5 qPCR machine. Flagged pools from the initial run were rerun on a second run as individual samples. Individuals associated with flagged barcodes were notified by 5 p.m. to begin isolation and to follow up with Health Services for diagnostic confirmation. This image was created by BioRender.com.

alternative to nasopharyngeal testing. For our RT-qPCR methodology, we adopted an approach similar to the FDA-EUA (Emergency Use Authorization) Yale SalivaDirect protocol (14). This technique does not require preservatives in the saliva collection tube or an RNA extraction step, and it detects both SARS-CoV-2 and human RNase P (RP) control in the same assay by duplexing of primer-probe sets. We utilized the validated approach that omits proteinase K treatment, replacing it with a cost-effective heat-inactivation step that was first demonstrated by researchers at the University of Illinois at Urbana-Champaign (UIUC) to be very effective (15).

Further, we used a pooled testing approach to maximize both throughput and cost savings, pooling 5 or 10 saliva samples per test. Watkins et al. investigated the effect of pooling saliva samples on RT-qPCR sensitivity and predicted by regression analysis that the loss of sensitivity by pooling is relatively small (7.4% lower for pools of 5 and 11.1% for pools of 10 [16]). In addition to substantial cost savings, the efficiency of our pooled testing approach enabled the rapid (same-day) turnaround of results with a single qPCR machine, facilitating immediate isolation of infectious individuals (Fig. 1).

At just $0.43/sample, our testing program identified 83% of all known COVID-19 cases on campus during the spring semester, most of which were from individuals who were experiencing mild or no symptoms at the time of testing. Importantly, routine testing in the spring reduced weekly COVID-19 case counts by 60%, allowing for a more relaxed campus experience, as students were able to visit other dorm rooms, quarantine less frequently, and participate in athletic competitions and student group activities. Thus, our findings support the use of routine, pooled SARS-CoV-2 surveillance testing for the rapid identification and isolation of infectious individuals to reduce virus spread.

## RESULTS

**Assay sensitivity and consistency.** We followed the Yale SalivaDirect assay, which uses duplex RT-qPCR to identify the SARS-CoV-2 nucleocapsid gene at region 1 (N1) for infected individuals and human RNaseP (RP) as a valid saliva sample (14). We achieved a limit of detection (LOD) of 500 copies/ml using gamma-irradiated SARS-CoV-2, reliably picking up 19/20 samples spiked at 500 and 1,000 copies/ml, and consistently detected virus over the full dilution range (Fig. 2A). We further contrived positives to mimic clinical validation according to FDA guidelines and again reliably detected positives both at 2× the LOD and across the assay range (Fig. 2B). To monitor assay precision,

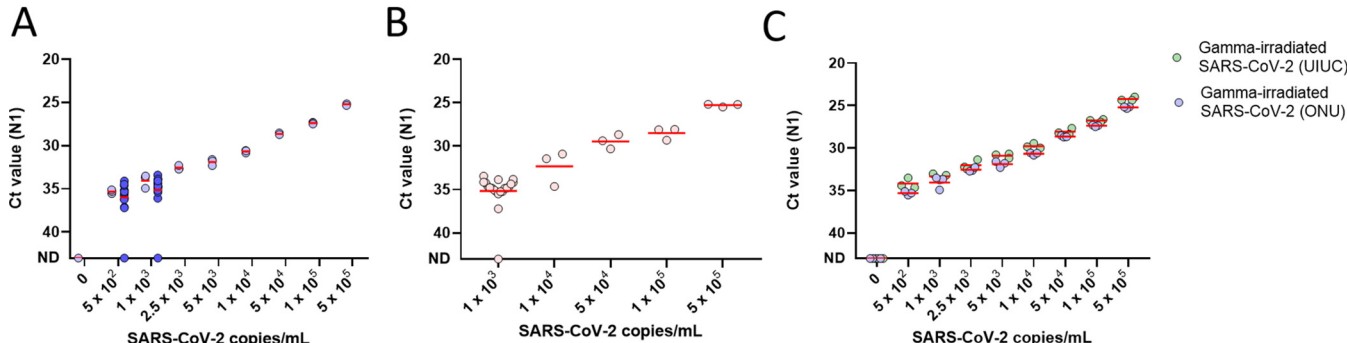

**FIG 2** Limit of SARS-CoV-2 detection. (A) Gamma-irradiated SARS-CoV-2 was spiked into saliva from a pool of negative individuals. After the initial serial dilution was run in triplicate (light circles), the limit of detection (LOD) was confirmed by testing 20 samples near the suspected LOD (dark circles, 19/20 detected at both 500 and 1,000 copies/ml). The red lines indicate the mean. ND indicates that the SARS-CoV-2 nucleocapsid gene region 1 (N1) was not detected. (B) We tested 30 additional contrived saliva specimens—20 at 2× the LOD and the rest spanning the range of the assay. Each contrived sample contained gamma-irradiated SARS-CoV-2 spiked into a unique, negative saliva sample. The red lines indicate the mean of replicates. (C) Our LOD compares favorably to the UIUC protocol. ONU's saliva protocol was compared with the UIUC protocol, which first dilutes saliva one-to-one with 2× TBE and then adds 0.5% Tween 20 after heat inactivation. The UIUC protocol uses 1.3× saliva sample per reaction compared to ONU's protocol. Red lines indicate the mean of three replicates.

we included a SARS-CoV-2-positive control on every run and achieved consistent N1 $C_T$ values (see Fig. S1A in the supplemental material). Consistent intra- and interrun N1 $C_T$ values were confirmed, further supporting assay precision (Fig. S1B and C).

The UIUC saliva protocol for SARS-CoV-2 detection included various stabilizing buffers, achieving the highest sensitivity by diluting saliva with 2× Tris-borate-EDTA (TBE) and then adding 0.5% Tween 20 after heat inactivation (15). Our simplified approach was comparable to the UIUC protocol (Fig. 2C). While the UIUC protocol did improve N1 detection by 1.1 cycles at the LOD, it should be noted that their assay uses more saliva (1.3×), which may account for the gain in signal. Lastly, we found that the quality of the saliva sample, as indicated by the RP $C_T$ value, neither correlated with nor influenced N1 $C_T$ values ($R^2 = 0.0036$; Fig. S1D).

**Pooling for rapid screening of an undergraduate population.** To achieve rapid screening of a residential population of undergraduates on a regular basis, we utilized a two-stage pooled approach to obtain same-day results for up to 930 samples (Fig. 1). Saliva samples were combined in pools of 5 or 10 for screening in the first run, and positive pools were subsequently deconvoluted in a second run. All residential students were required to submit weekly saliva samples, and residential and commuter students that were part of a higher exposure group (athletics, music ensembles, theater, etc.) were required to submit saliva samples twice a week (Monday/Thursday or Tuesday/Friday; Table S2).

Collectively, we administered 43,884 saliva tests during the semester, flagging 114 (83%) of 138 total positives during the spring semester (Table 1). Notably, classes resumed

**TABLE 1** Campus testing numbers

| SARS-CoV-2 results, spring 2021 | Total | Positives (%) |
|---|---|---|
| Saliva tests administered | 43,884 | |
| Intake testing (Jan 14–18) | 2,163 | |
| Semester testing (Jan 19–May 7) | 41,721 | |
| Saliva SARS-CoV-2 positive | 114 | 83 |
| Confirmed positive | 109 | |
| Intake testing | 20 | (14) |
| Semester testing | 89 | (64) |
| Antigen negative, no sign of current infection | 5 | (4) |
| Intake testing | 2 | |
| Semester testing | 3 | |
| Health services (nonsaliva) tests[a] | 205 | |
| Positive | 24 | 17 |
| Negative | 180 | |

[a]Some nonsaliva positives came from outside Health Services.

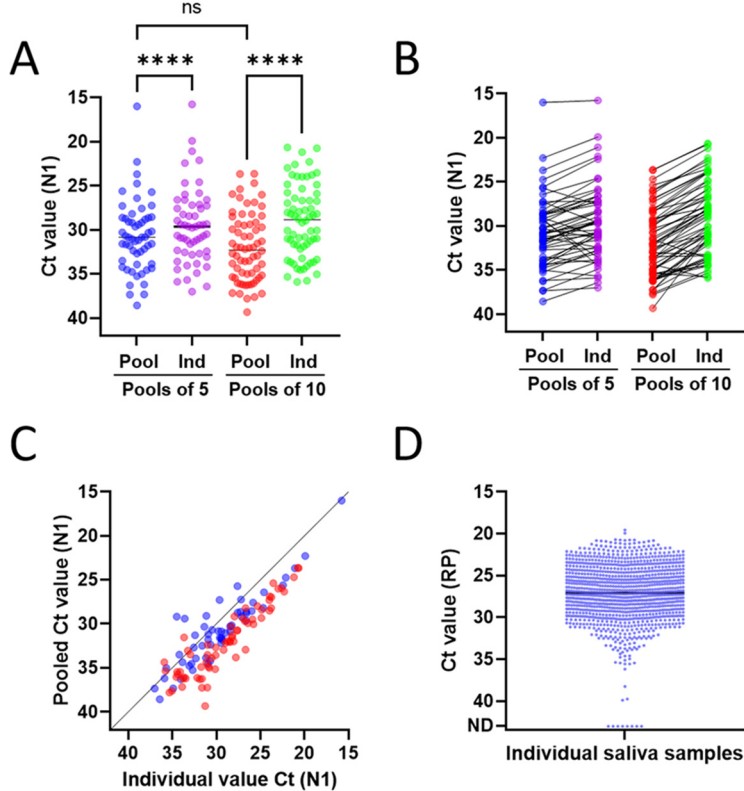

**FIG 3** Pooled and individual detection of SARS-CoV-2 in human saliva. (A to C) Comparison of N1 $C_T$ values for saliva samples from pools of 5 and 10 in which SARS-CoV-2 was detected versus the deconvoluted, individually flagged sample(s) from each pool. (A) Mean and (B) paired samples over the assay range. Pools of 5 had a mean $C_T$ value difference from individuals of 1.18 (95% CI [0.69, 1.68]). Pools of 10 had a mean $C_T$ value difference from individuals of 3.05 (95% CI [2.63, 3.47]). (C) Pooled versus individual $C_T$ values for N1 demonstrate consistency of signal detection over the assay range. Pools of 5 are shown in blue, and pools of 10 are shown in red. Pools of 5, Pearson's $r(55) = 0.9038$, $P < 0.0001$. Pools of 10, Pearson's $r(66) = 0.9108$, $P < 0.0001$. (D) RP $C_T$ values for one replicate of each individual saliva sample analyzed on the deconvolution runs throughout the semester; this includes N1-negative individuals, which were the majority of deconvoluted samples. ****, $P \leq 0.0001$; ns, not significant as determined using a Holm-Šídák multiple-comparison test.

at a time (19 January 2021) of heightened outbreak in the United States and locally in the Midwest. Despite the enhanced testing during the spring semester, the average number of new cases per week post intake was only 7.3 in the spring versus 18.2 in the fall. Of the confirmed positives, intake saliva testing identified 20 fever-free, asymptomatic individuals (14% of semester positives) during move-in before significant spread could occur on campus. After intake, saliva testing flagged 89 individuals over the remaining semester.

We processed an average of 665 saliva samples per testing day. To maximize assay sensitivity, we filled 96-well plates with as many pools of 5 as possible with the remainder in pools of 10. As shown in Fig. 3A, pools of 5 and 10 displayed a loss of sensitivity of 1.2 and 3.0 $C_T$, respectively, for N1 (95% CIs [0.69, 1.68] and [2.36, 3.47]). Based on our LOD findings in Fig. 2A, this would correspond to raising the input threshold to 1,000 to 2,500 copies/ml when these samples are diluted in pools of 5 and 10, respectively, compared to 500 copies/ml for samples run individually. These values are still below the EUA LOD of 6,000 to 12,000 copies/ml used in the Yale SalivaDirect protocol (14) and still over 2 to 3 orders of magnitude more sensitive than EUA rapid antigen tests (17, 18).

Pooling generally resulted in higher N1 $C_T$ values than when samples were run individually in the deconvolution run (Fig. 3B), though several did not follow this trend, suggesting that saliva may not contain homogenous viral sample. Notably, the relationship between sensitivity loss in pooled and deconvoluted runs was consistent and independent of the N1 cycle threshold, indicating similar detection reliability across the assay range (Fig. 3C). Moreover, 63% and

**Table 2** Follow-up deconvolution detection agreement

| Deconvolution replicates | Avg N1 $C_T$ | One detected (%) | Replicates detected (%) | |
|---|---|---|---|---|
| | | | Two | Three |
| Singlet | 28.62 | 19/19 (100) | NA[a] | NA |
| Duplicate | 30.41 | 43/43 (100) | 42/43 (98)[b] | NA |
| Triplicate[c] | 34.15 | 36/39 (92) | 36/39 (92) | 34/39 (82) |

[a]NA, not applicable.
[b]The $C_T$ value for pool with undetected duplicate was 36.27.
[c]Three false-positive pools occurred during semester testing with an average $C_T$ value of 36.24. Two of these occurred during opening week and may have been contamination.

33% of positives came from pools of 10 and 5, which is congruent with expectations, as pools of 10 and 5 comprised 67% and 30% of total tests, respectively (Table S3).

A potential concern for our approach was that after intake testing, students submitted saliva samples on their honor without any monitoring of sample collection. However, of the 1,091 individuals assayed during a deconvolution run, only 8 came back without RP signal, indicating that 99.3% were valid saliva samples (Fig. 3D). Furthermore, participation was strong, with 92% of assigned tests submitted successfully. Most missed tests came from students missing 1 to 2 tests, with only 3.8% from students missing 3 or more assigned days (data not shown).

When we could accommodate replicates in our deconvoluting run, individuals from pools with higher N1 $C_T$ values were run in duplicate, and those with the highest $C_T$ values were run in triplicate, in order to ensure correct identification and assay sensitivity in pools with lower viral load (higher N1 $C_T$ values). Our data suggest that running deconvolution runs in singlet from pools with $C_T$ values below 35 is sufficient for reliable deconvolution detection. Running deconvolution samples in duplicate could continue to ensure correct identification when the pooled $C_T$ value for N1 is above 35 (Table 2).

**COVID-19 survey.** Consistent with having 77% (1,576 of 2,035) of all student saliva testers consent to research, 73% of the flagged samples throughout the semester were from students consenting to the research study. Our survey response rate was 76%, resulting in the completion of 66 surveys—14 who were identified during intake testing (21%) and 52 during the semester (79%). All respondents were between the ages of 18 and 25, and 62.1% were female (which corresponds with 56.6% of the student body being female). Residential students comprised 97% of respondents, and 81.8% lacked preexisting health conditions (additional details in Table S4).

Of the student respondents whose saliva was flagged, 92.4% received diagnostic confirmation of SARS-CoV-2 by an FDA-EUA test. Of the five (7.6%) who did not receive diagnostic confirmation, three were flagged upon intake testing or early in the semester and demonstrated evidence of recent, undiagnosed infection, so persistent viral shedding is the presumed source of the saliva results. Prior health history of the remaining two is unknown; one individual experienced common cold symptoms in the 10 days after their saliva test, while the other remained asymptomatic.

Of the 61 student respondents who received diagnostic confirmation of COVID-19, 12 tested negative on their first diagnostic test but later tested positive on a second test. Of the 51 students who specified the provider of their diagnostic test(s), 84% noted that their testing was done by the campus's Health Services department, whose routine was to administer an Abbott BinaxNOW rapid antigen test within 1 to 2 days of the flagged saliva test and then the same test again 48 h from the first test if it was negative. The mean N1 $C_T$ value of those who first tested negative was significantly higher than that of those whose first test was positive (31.18 versus 28.51, respectively; $P = 0.0234$) (Fig. 4A). The first diagnostic test of those with an N1 $C_T$ value of $<30$ for their saliva test was consistent 90.6% of the time, as opposed to only 69.0% for those with an N1 $C_T$ value of $\geq30$ (Table S5).

Of those with a diagnostically confirmed COVID-19 case and reported symptoms, 36.2% (21/58) reported being asymptomatic at the time of their flagged saliva test, 48.3% (28/58) reported very mild symptoms, and 15.5% (9/58) reported very noticeable

Microbiology Spectrum

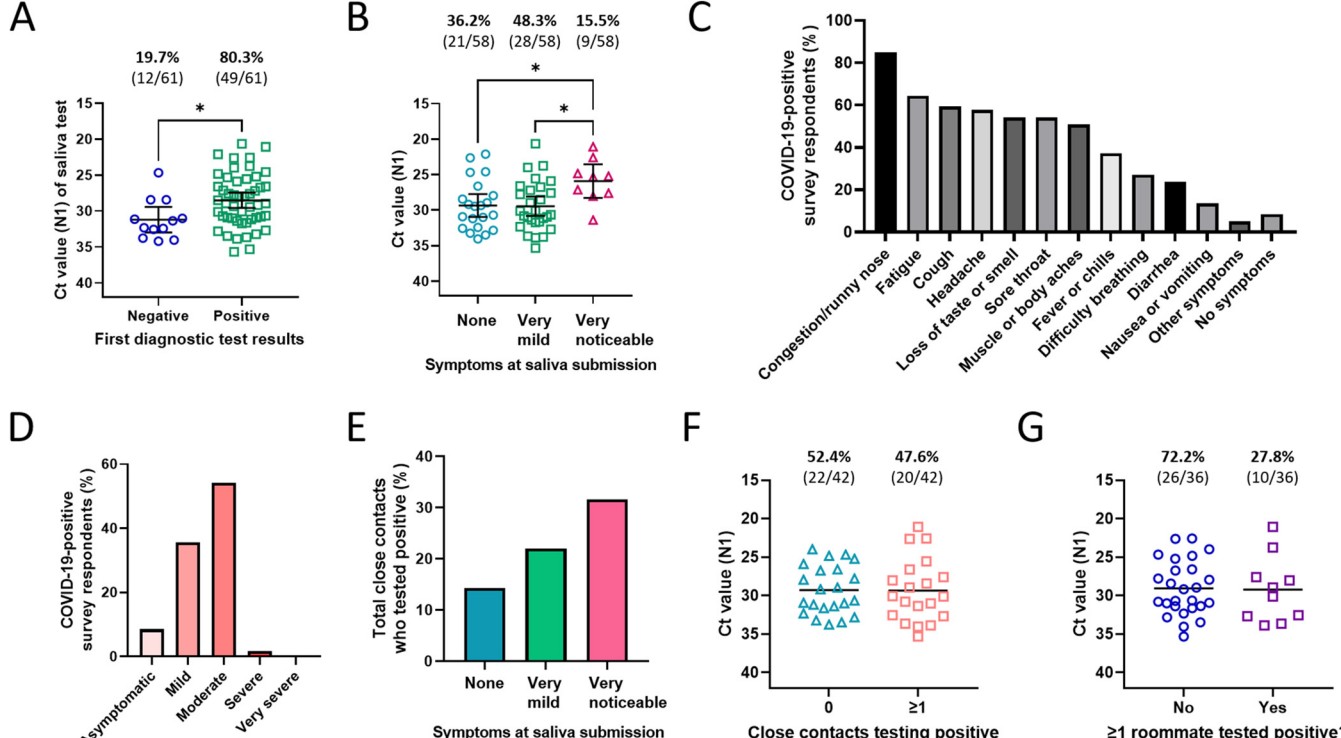

**FIG 4** Diagnostic outcomes, symptoms, and contagiousness of COVID-19-positive survey respondents. (A) Comparison of N1 $C_T$ values of COVID-positive survey respondents whose first diagnostic test result was negative versus positive. Error bars represent the mean with 95% confidence interval (CI). *, = $P \leq 0.05$ by a two-tailed, unpaired $t$ test. (B) Comparison of N1 $C_T$ values of students reporting various degrees of symptoms at the time of their flagged saliva test. Error bars represent the mean with 95% CI. *, $P \leq 0.05$ in *post hoc* Tukey's multiple-comparisons tests. (C) Symptoms experienced by COVID-19-positive survey respondents in the 1 to 2 weeks after their flagged saliva test. "Other symptoms" reported included rash, dizziness, and bruising. (D) Severity of COVID-19 symptoms for survey respondents during the 1 to 2 weeks after the flagged saliva test. Mild symptoms were defined as minor discomfort such as loss of taste/smell or like a common cold; moderate symptoms were defined as flu-like symptoms that were managed at home; severe symptoms required treatment from a doctor; and very severe symptoms required hospitalization. (E) Total (collective) percentage of close contacts who tested positive within 1 to 2 weeks after exposure to a survey respondent, categorized by symptom status of the survey respondents at the time of the flagged sample submission. The number of COVID-19-positive close contacts out of total close contacts reported are as follows: 7/49 reported by respondents without symptoms, 18/82 by those with very mild symptoms, and 6/19 by those with very noticeable symptoms. (F) Comparison of N1 $C_T$ values of survey respondents who had no or at least one close contact test positive in the 1 to 2 weeks after the respondent's saliva was flagged. Black lines represent the mean. (G) Comparison of N1 $C_T$ values of survey respondents who did or did not have at least one roommate test positive for COVID-19 in the 1 to 2 weeks after the respondent's saliva was flagged. Black lines represent the mean.

symptoms. Students experiencing very noticeable symptoms at the time of the flagged saliva test had a significantly lower $C_T$ value than those with no or very mild symptoms ($P = 0.0405$ and $0.0266$, respectively), indicating higher viral load in their saliva (Fig. 4B). For those not experiencing symptoms at the time of the flagged test but who recalled when symptoms later developed ($n = 12$), the symptoms developed an average of 1.7 days (median of 2 days) after the flagged test day. For those already experiencing symptoms at the time of their flagged test and who recalled when symptoms developed ($n = 30$), the symptoms developed an average of 1.6 days (median of 1 day) prior to the flagged test day (data not shown).

The most common symptoms among all COVID-19-positive respondents were congestion or runny nose (84.7%), fatigue (64.4%), and cough (59.3%) (Fig. 4C). One of the few distinguishing symptoms of COVID-19, loss of taste or smell, was experienced by slightly more than half (54.2%) of individuals, and fever or chills was experienced by only slightly more than a third (37.3%). The average number of symptoms reported by an individual was 5.3 symptoms. Symptoms were mild (defined as minor discomfort such as loss of taste/smell or like a common cold) for 35.6% of students, moderate (flu-like symptoms that were managed at home) for 54.2%, or severe (required treatment from a doctor) for 1.7% (Fig. 4D). No students reported very severe symptoms (defined as those requiring hospitalization), and only 8.5% of students remained asymptomatic throughout the window of infection. None of the above-described students had been

partially or fully vaccinated, none had previously been known to have COVID-19 antibodies in their blood, and only one (1.7%) had previously tested positive for COVID-19 at least 3 months prior, suggesting a repeat infection.

Finally, we investigated the contagiousness of individuals who tested positive during the semester (excluding intake testing) and reported close contacts ($n = 42$). A close contact was defined as someone within 6 feet of distance for at least 15 (cumulative) min within 48 h of submitting the flagged sample. In total, 150 close contacts were reported by these 42 student respondents, and 31 of those close contacts (20.7%) were reported to have since tested positive for COVID-19. Respondents with at least one close contact who tested positive since the respondent tested positive amounted to 47.6%. Students experiencing very noticeable symptoms at the time of the flagged saliva test were twice as likely to have a close contact test positive than those experiencing no symptoms (Fig. 4E). Of the 36 who had a close-contact roommate, 27.8% had one or more roommates subsequently test positive. Interestingly, the N1 $C_T$ value of individuals who had at least one close contact or roommate test positive was not significantly different than that of those who did not ($P = 0.9502$ and 0.9108, respectively) (Fig. 4F and G), and no significant correlation was identified between an individual's N1 $C_T$ value and the percentage of their close contacts who later tested positive for COVID-19 ($R^2 = 0.0085$; $P = 0.5621$).

## DISCUSSION

Routine screening of populations during the COVID-19 pandemic that is rapid, highly sensitive, and cost-effective has been elusive. To date, financially privileged organizations such as professional sports have primarily been the ones to adopt routine screening at significant cost in order to remain open (19, 20). Here, we demonstrate the implementation of a 2-stage, pooled saliva testing strategy that offers same-day results in under 8 h, is among the most sensitive tests available with detection of virus as low as 2,500 copies/ml in pools of 10, and is cost-effective at under $1 per sample. This approach allowed for successful screening of 43,884 samples and detection of 83% of the semester's COVID-19 cases on a residential college campus.

**A two-stage pooling approach cuts down on time and cost and reduces consumable waste.** When routine testing is not feasible due to supply chain issues or cost, many universities elect to use either random testing or surveillance-based informative testing (SBIT), where random surveillance testing is used to identify outbreaks, with follow-up testing of hot spots. While SBIT offers a 24% reduction in cases over random testing alone, modeling data also indicate that implementing weekly and twice-weekly testing of all undergraduates would result in 36% and 72% fewer infections, respectively, compared with SBIT (3). Assay sensitivity also improves efficacy when modeling a college campus, as weekly testing using highly sensitive molecular tests such as RT-qPCR is predicted to reduce over 60% of total infectious individuals compared with just under 50% using point-of-care tests. Importantly, twice-weekly testing nearly completely abrogates all modeled case spread regardless of assay sensitivity. In terms of infection on a campus of 20,000 students, this is the reduction of outbreaks of nearly 2,000 cases without screening (only symptom monitoring), to just over 200 with weekly testing, to under 20 with twice-weekly testing (21).

While routine testing with high sensitivity is ideal, Larremore et al. found that result turnaround time is more important than assay sensitivity for isolating infectious individuals and limiting spread. In their model, testing every 3 days could abrogate by nearly 100% the total number of infectious individuals if results were received on the same day, but infectious individuals were reduced by only approximately 70% if test results were delayed by 2 days. Weekly testing reduced the number of infectious individuals by only 20%, demonstrating that the advantage of routine screening is lost without the timely turnaround of test results (21).

The cost of routine testing has been a major deterrent. The Yale SalivaDirect protocol is one of the least expensive SARS-CoV-2 assays available, with reagent costs ranging from $1.21 to $4.29 per sample (14). However, these cost estimates only account for assay chemistry and do not account for plates, pipette tips, or other consumables. When we include all consumables from saliva collection through processing, our individual sample cost is $2.09

(Table S1). However, pooling improves these costs considerably. We averaged 665 tests per day, which brought our average per sample cost to $0.43 and reduced consumable waste at a time when global supply chain issues are common. Moreover, our pooling strategy allowed for a turnaround time of 8 h, which could further be improved upon with laboratory automation. In contrast, running our samples individually without additional technicians or qPCR equipment would have required seven individual plates spaced over 2 days and would have prohibited twice-weekly testing. In other words, pooling compounds savings and reliability when labor, equipment, and supplies are factored in.

The flexibility of this approach offers scalability to larger populations and can be tailored to SARS-CoV-2 positivity rates. Pools of 10 still picked up viral load at 2,500 copies/ml, and pools of 5 at 1,000 copies/ml, demonstrating minimal loss of signal and preserving sensitivity at over 2 to 3 orders of magnitude better, respectively, than rapid antigen tests (17, 18). Cost and time efficiency are maximized in our format when case positivity is below 1%, where the maximum per sample cost in pools of 10 is $0.47 with 9 positive pools. Pools of five are recommended when case positivity is greater than 1% but still less than 3.9%. At 3.9%, pools of 5 increase the maximum per sample cost to $0.87 with 18 positives, and beyond 3.9% case positivity (>18 positives per 465 individuals with our format), the utility of a 2-staged approach diminishes. This is in agreement with a modeled analysis that revealed that for a prevalence above 2.6%, pools of 5 are most efficient, whereas below 2.6%, pools of 10 or 20 are most efficient in terms of total tests run (16). It should be noted that pooling does increase the complexity of running the assay, though errors can be minimized with organized workflow and procedural checks by a lab manager.

Several universities have developed in-house COVID-19 tests for routine surveillance testing of their student populations (3, 14, 15). While in-house testing does present a rather significant cost for universities, the benefit of remaining open for on-campus instruction likely outweighs this cost. For example, Clemson University estimated that in-house, weekly testing would cost approximately $44,000/week. However, over the semester, the university generated approximately $1.65 million/week in housing and dining revenue, which would have been lost if outbreaks forced campus closure (3). Thus, lost revenue must also be considered when assessing the cost of routine testing.

**Saliva as a model medium for surveillance testing.** Several studies have demonstrated high agreement between saliva and nasopharyngeal samples (11, 12), and in some cases, saliva is superior for SARS-CoV-2 detection (13, 22). Moreover, saliva sampling is noninvasive, does not require transport medium, and is stable at room temperature for at least 24 h (15), exhibiting minimal degradation over 7 days (14). With a few educational videos, posters, and monitoring of their first submission, we permitted self-collection using the honor system for the entirety of the semester and collected 99.3% valid samples, and 92% submitted on their assigned dates. Moreover, supporting the superior sensitivity of our pooled testing approach, 19.7% of our flagged positives were not confirmed as positive with an initial rapid antigen test (typically Abbott BinaxNOW from our Health Services office) but did confirm as positive with a second test 24 to 48 h later. In fact, our agreement of 80.3% on a first confirmatory test is 27% higher than what others have reported for college students (23) and likely reflects the fact that in our hands, confirmatory tests were typically administered 24 to 48 h after saliva submission, allowing viral load to increase sufficiently for antigen detection.

**Preventing unknowing spread of SARS-CoV-2 on a traditional undergraduate campus.** Congregate living among 18- to 22-year-olds presents its own set of unique challenges, as they have fewer severe symptoms and thus may be less likely to seek out symptomatic testing. Indeed, others have reported 51% of college students to be asymptomatic at the time of screened testing (2). In our study, 36.2% were asymptomatic, but we also found that 48.3% reported experiencing very mild symptoms at the time of testing. Combined, this suggests that 84.5% of college-aged students may not have sought out a SARS-CoV-2 test on their own at the time of screening. Moreover, only 56% developed moderate (flu-like) or severe symptoms over the course of their illness. The key distinguishing symptom of COVID-19, loss of taste or smell, was experienced by 57% of mild (cold-like)

self-reporters. The remaining asymptomatic and mild cases comprise 24% of all positive cases in this study, who likely would not have sought out a symptomatic test at any point.

A systematic review of literature found that loss of taste and smell occurs on average 4 to 5 days after the onset of other symptoms (24). This suggests that of our mild cases that did go on to lose taste and smell, we likely flagged them as positive well before this key distinguishing symptom of COVID-19 developed. Furthermore, those that did not lose taste and smell but had mild symptoms likely would not have sought out testing. Just based on asymptomatic and mild cases, we estimate that we may have removed 348 days of infectious individuals unknowingly spreading the virus (9 cases asymptomatic throughout × 10 days of isolation + 22 mild cases × 4 days before loss of taste or smell would prompt them to seek symptomatic testing + 17 mild cases that never lost taste and smell × 10 days of isolation). Added to this, we likely achieved isolation of the 56% that went on to develop flu-like symptoms or worse well before they would have sought out symptomatic testing. Reducing this time is particularly important, as social distancing and masking adherence are challenging in athletic or other student groups, and especially in college dorms and apartments.

The value of identifying positives as early as possible was highlighted further by our observation that the degree of symptoms experienced at the time of testing was predictive of the number of close contacts that became positive (Fig. 4E). Interestingly, $C_T$ values were not indicative of infectiousness, as $C_T$ values were virtually identical for those that did or did not have close contacts who became positive, including when analysis was restricted to roommates. Moreover, we observed a nearly identical phenomenon when we compared data for once and twice per week testers (data not shown). It should be noted that with much community spread, it is difficult to determine definitively if these developed as a result of, or parallel to, the surveyed students. Regardless, these data highlight the challenges of relying solely on symptomatic individuals to seek out testing, as well as those inherent in congregate living.

Indeed, in the fall, we relied solely on self-monitoring and symptomatic individuals seeking a SARS-CoV-2 test. This resulted in 273 known cases and a much more restrictive campus experience, including reduced student groups/activities, takeout-only dining during outbreaks, heightened policing of mask and distancing guidelines, and the inability to visit other dorm rooms or apartments. Conversely, screening in the spring semester allowed for open dorms, open dining halls, all athletic teams competing throughout, and no mitigation communications. With high compliance of saliva testing, the 138 total positives likely represent the extent of spring semester spread. In contrast, since we have found that 24% are asymptomatic or mildly symptomatic without loss of taste or smell throughout their illness, it is likely that in the fall, our 273 known cases missed an additional 86 cases that never became ill enough to seek testing. It should be noted that this inference relies heavily on self-reported survey data and thus may not represent as accurate a picture as data from a health clinic. However, our large participation number and rate suggests that our data are a reasonable reflection of SARS-CoV-2 spread among the college-age demographic.

Together, our data support the need for widespread testing that is rapid, highly sensitive, and cost-effective. As the pandemic moves to secondary phases of increased openness, but with lagging vaccination rates among undergraduate populations (1), routine testing remains one of the best options for minimizing viral spread while maintaining as much normalcy as possible. Organizations such as colleges cannot reasonably create a bubble and are thus susceptible to new cases coming onto campus or going out into the community. Pooled saliva testing such as ours that maintains high sensitivity and rapid turnaround should receive strong consideration going forward so that routine screening is normative and not reserved for professional sports or other elite institutions.

## MATERIALS AND METHODS

**Project design, setting, and population.** Spring semester (14 January to 7 May 2021) surveillance testing was conducted on ONU's Bourbonnais campus in Illinois, USA, and was required for residential students. Students participating in organized group activities were classified as higher-exposure and

tested twice weekly, while all other undergraduates were tested weekly. Missed tests resulted in temporary restriction from group activities, and fines were issued upon third and subsequent misses.

Upon arrival, students were instructed to collect 0.5 ml of naturally pooling saliva into 1.5-ml microcentrifuge tubes and were advised not to eat, drink, or conduct dental hygiene for 30 min prior to collection. After intake, samples were deposited in drop-off bins as early as 2:00 p.m. the afternoon before their assigned day and collected at 9:00 a.m. each testing day.

Additionally, university policy was to wear masks while indoors (except while in one's own dorm room or family unit), maintain six feet of distancing indoors, limit event and class sizes to 50, isolate students who tested positive for SARS-CoV-2, and quarantine for 10 days for individuals in close contact with anyone testing positive for COVID-19, according to CDC guidance. Prior to returning to campus in the fall and spring, all students were asked to voluntarily self-quarantine for 14 days.

**SARS-CoV-2 detection. Assay performance validation.** The limit of detection (LOD) was determined using gamma-irradiated SARS-CoV-2 obtained through BEI Resources, NIAID, NIH (SARS-CoV-2, isolate USA-WA1/2020, gamma-irradiated, NR-52287, contributed by the Centers for Disease Control and Prevention). Gamma-irradiated SARS-CoV-2 was spiked at 0 to 50,000 copies/ml into pooled saliva and assessed by RT-qPCR. We subsequently contrived samples at the suspected LOD to determine the concentration at which 19 out of 20 contrived samples would be detected. To confirm assay performance, we tested 30 additional contrived unique saliva specimens—20 at 2× the LOD and the rest spanning the range of the assay. We considered 95% agreement at 2× the LOD and 100% agreement at all other concentrations as acceptable accuracy.

Intrarun and interrun precision were assessed from replicates on a given plate or over multiple daily runs, respectively, using saliva from individuals with diagnostically confirmed SARS-CoV-2 infection. For intrarun precision, 10 replicates each of 3 positive saliva samples were tested in a single run. Two replicates each of three positive saliva samples were tested daily for 7 days for interrun precision. For the latter, samples were aliquoted and frozen in individual tubes to be thawed on each test day to ensure uniform sample quality across days.

**RT-qPCR for SARS-CoV-2.** We closely followed the SalivaDirect protocol of Vogels et al. without proteinase K digestion (14) and modified it for a two-stage pooled approach. Briefly, all saliva samples were kept at room temperature and processed on the same day of collection in a biosafety level 2 (BSL-2) laboratory. Samples were pooled in sets of 5 or 10 into a total volume of 50 $\mu$l, placed into a thermocycler (Labnet, TC9600-G), and heat-treated for 30 min at 95℃. Duplex PCR utilized the CDC2019-nCoV (N1) and human RP control primer-probe sets (Integrated DNA Technologies; Table S1).

Assay mix—3.0 $\mu$l primer-probe mix (400/200 nM primer/probe for N1 and 150/200 nM primer/probe for RP), 3.75 $\mu$l TaqPath 1-Step RT-qPCR Master Mix, CG (Fisher Scientific), and 4.5 $\mu$l nuclease-free water—was aliquoted into each well of a 96-well plate. Then, 3.75 $\mu$l of heat-treated saliva was added for a final reaction volume of 15 $\mu$l, which conserves reagents at three-fourths the reaction volume used by Vogels et al. Each plate included two negative controls (nuclease-free water) and one positive control (2019-nCoV_N_Positive Control [Integrated DNA Technologies]).

Targets (N1-FAM and RP-Cy5) were assessed on a QuantStudio5 qPCR system (Life Technologies Corp.). qPCR parameters were as follows: 52℃ for 10 min, 95℃ for 2 min, and then 40 cycles of 10 sec at 95℃ and 30 sec at 55℃. Positive pools (N1 $C_T$ values below 40) were deconvoluted on a second run as individual samples.

**Surveillance outcomes.** Given the urgency and rapid timeline for development of our testing program, we opted to use a surveillance testing strategy that did not diagnose COVID-19 but flagged potentially positive individuals and referred them for diagnostic COVID-19 testing with an FDA-EUA test; no patient-specific COVID-19 results were directly reported by our lab to state or local public health departments. Flagged positives were immediately notified and referred for diagnostic follow-up, and the student was moved to isolation housing until diagnostic results were obtained. If the student was diagnosed with COVID-19, he/she was allowed to return to regular housing and in-person activities 10 days after symptom onset or the date of the flagged saliva test, whichever was earlier per CDC isolation guidelines. Contact tracing was utilized for each case as described above.

**COVID-19 research survey.** Individuals 18 years and older participating in the SARS-CoV-2 surveillance testing could voluntarily participate in a survey-based research study, which was approved by ONU's Institutional Review Board. To incentivize students to participate in the research study, we offered 10 $25 gift card drawings for those who consented. Any consenting individual whose saliva was flagged throughout the semester was emailed the survey 8 to 11 days after virus detection (supplemental materials survey questionnaire). Two of the 68 completed surveys were by faculty/staff, but for consistency of the study population, we report only data from the 66 students.

**Data availability.** The data sets used and/or analyzed during the current study are available from the corresponding authors upon reasonable request.

**Figures and statistical analyses.** Figures were generated using GraphPad Prism 9 (version 9.1.2). Statistical analyses were also performed using this software, with a $P$ value of $\leq$0.05 considered statistically significant.

## SUPPLEMENTAL MATERIAL

Supplemental material is available online only.
**SUPPLEMENTAL FILE 1**, PDF file, 0.5 MB.

## ACKNOWLEDGMENTS

We thank two additional COVID-19 testing technicians, Gretchen Brinkman and Noah Finney, for their hard work throughout the semester. We thank Emily Wirick at Pro-Testing

Solutions, LLC, for optimizing our COVID-19 testing protocol. We thank Jennifer Chase at Northwest Nazarene University for sharing COVID-19 testing strategies and resources with our department to help make our testing a success. We thank the many members across ONU's campus that made the testing possible and successful during the spring semester, including members of Public Safety, Residential Life, and Health Services teams. We thank Jori Sharda for critical review and editing of this manuscript.

This research study was approved by Olivet Nazarene University's Institutional Review Board. Participants voluntarily signed an informed consent form if they wished to participate in the study.

This research was made possible by the support of Olivet Nazarene University.

Authors' contributions will be provided upon request.

We declare no conflicts of interest.

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
