## [Reviewer comments · Microbiology Spectrum]

Microbiology Spectrum

Routine, Cost-effective SARS-CoV-2 Surveillance Testing Using Pooled Saliva Limits Viral Spread on a Residential College Campus

Nicole Vander Schaaf, Anthony Fund, Brianna Munnich, Alexi Zastrow, Erin Fund, Tanner Senti, Abigail Lynn, Jonathon Kane, Jennifer Love, Gregory Long, Nick Troendle, and Daniel Sharda

Corresponding Author(s): Daniel Sharda, Olivet Nazarene University

Review Timeline:

Submission Date:	July 29, 2021
Editorial Decision:	September 6, 2021
Revision Received:	September 21, 2021
Accepted:	September 21, 2021

Editor: Tulip Jhaveri

Reviewer(s): Disclosure of reviewer identity is with reference to reviewer comments included in decision letter(s). The following individuals involved in review of your submission have agreed to reveal their identity: Luciana Jesus Costa (Reviewer #1); Akshat Mullerpatan (Reviewer #2)

Transaction Report:

DOI: <https://doi.org/10.1128/Spectrum.01089-21>

September 6, 2021

Dr. Daniel R Sharda
Olivet Nazarene University
Biological Sciences
One University Ave
Bourbonnais, IL 60914

Re: Spectrum01089-21 (Routine, Cost-effective SARS-CoV-2 Surveillance Testing Using Pooled Saliva Limits Viral Spread on a Residential College Campus)

Dear Dr. Daniel R Sharda:

Thank you for submitting your manuscript to Microbiology Spectrum. When submitting the revised version of your paper, please provide (1) point-by-point responses to the issues raised by the reviewers as file type "Response to Reviewers," not in your cover letter, and (2) a PDF file that indicates the changes from the original submission (by highlighting or underlining the changes) as file type "Marked Up Manuscript - For Review Only". Please use this link to submit your revised manuscript - we strongly recommend that you submit your paper within the next 60 days or reach out to me. Detailed information on submitting your revised paper are below.

Link Not Available

Sincerely,

Tulip Jhaveri

Journals Department
Reviewer comments:

Reviewer #1 (Comments for the Author):

In their manuscript entitled: " Routine, Cost-effective SARS-CoV-2 Surveillance Testing Using Pooled Saliva Limits Viral Spread on a Residential College Campus", Schaaf and co-workers undoubtedly demonstrated the usefulness and the reduced cost and turnaround time for results with their testing strategy in a College Campus student community. Moreover, the authors also contributed with relevant epidemiological information on this specific population , demonstrating the most frequent symptoms, the average of symptoms, and the percentage of asymptomatic individuals occurring in this population. The authors comprehensively demonstrated a high sensitivity of their saliva pooling approach for the standard RT-qPCR assay, The study is highly relevant given the actual epidemiological scenario and also very weel designed and conducted. The manuscript is weel written and results are presented clearly. Conclusions are in agreement with all results presented.

Reviewer #2 (Comments for the Author):

Overall, the paper presents a greatly useful case study for non-invasive Covid testing, employing only patient's saliva for the test. The studies are well-structured, and the appropriate controls have been performed within the constraints of student population and pandemic regulations. The data is presented in a clear manner and conclusions are well-substantiated with suitable statistical analyses. The discussion part of the manuscript captures several important points and explains key observations - authors have addressed the ease, effectiveness and economics of this technique, while also acknowledging potential pitfalls.

The paper is interesting and demonstrates great use of this testing procedure for quick result turnaround. However, a few minor points require explanation. Please see the attachment for specific comments.

Reviewer #3 (Comments for the Author):

The authors present a comprehensive body of evidence surrounding both prevention and mitigation of SARS-CoV-2 outbreaks on college campuses. Their adaption of a saliva-based assay to provide cost-effective testing with timely reporting is impressive. The findings they have on symptoms are valuable to the research community. I only have minor comments to help improve the clarity of the manuscript to help better highlight the impactful study that this is.

Major comments

The background is very thorough and almost reads as a complete study report. Therefore, I think to keep this as an introductory section, it could be work moving some key findings to the discussion, such lines 122 or 123-130 and lines 133-140. This would allow the authors to more clearly highlight the relevant background to their study and lead into what they did in response to this.

Figure 2 - why are the individuals only presumed negative? Were the samples tested prior to pooling to determine negative status?

Minor comments

Line 29 - suggest 'Limit of detection' rather than detection limit.

Line 109 - 'FDA-authorized' (only has EUA, not full approval)

Line 185 - 3.75 ul of lysate is a deviation from the SalivaDirect protocol and could be mentioned here.

Line 196 - 'by an FDA EUA (Emergency Use Authorization) test'

Line 227 - the authors could also consider stating 'the highest Ct values representing weakly positive samples' (or similar) for readers less familiar with RT-qPCR

Figure 1 - suggest 'RT-qPCR' for consistency with the text

Staff Comments:

Preparing Revision Guidelines

For complete guidelines on revision requirements, please see the journal Submission and Review Process requirements at <https://journals.asm.org/journal/Spectrum/submission-review-process>.

Submissions of a paper that does not conform to Microbiology Spectrum guidelines will delay acceptance of your manuscript. "

Please return the manuscript within 60 days; if you cannot complete the modification within this time period, please contact me. If you do not wish to modify the manuscript and prefer to submit it to another journal, please notify me of your decision immediately so that the manuscript may be formally withdrawn from consideration by Microbiology Spectrum.

Summary

Dear authors and editors,

Schaaf and co-workers present a study piloting a saliva-based COVID-19 testing procedure at the Olivet Nazarene University to screen for positive cases. The key findings of this paper include – adopting inexpensive pooling techniques to effectively and quickly enable the detection of SARS-CoV-2. The authors performed controls to identify the limit of detection at 500 copies/mL for individual samples and 1000-2500 copies/mL for pools of 5 and 10 samples, making it a very sensitive assay amenable to quick determination through pooling. 83% of the total positives obtained in the spring semester were through saliva-testing, which resulted in nearly a 60% reduction in the number of weekly cases and potentially 348 lost days due to illness/quarantine.

Overall, the paper presents a greatly useful case study for non-invasive Covid testing, employing only patient's saliva for the test. The studies are well-structured, and the appropriate controls have been performed within the constraints of student population and pandemic regulations. The data is presented in a clear manner and conclusions are well-substantiated with suitable statistical analyses. The discussion part of the manuscript captures several important points and explains key observations – authors have addressed the ease, effectiveness and economics of this technique, while also acknowledging potential pitfalls.

The paper is interesting and demonstrates great use of this testing procedure for quick result turnaround. However, a few minor points require explanation. Please see below for specific comments.

Major points: None

Minor points:

1. Abstract: None
2. Background: None
3. Materials and Methods:
 - Lines 171-174: Is it ok to run inter-run studies on different days? How does sample stability play a role and when is the sample gamma-inactivated if it's performed on a different day?
4. Results:
 - General comment: It would be good to mention upfront that there were 138 positives and what the split was between saliva and non-saliva tests to better orient the reader.

- Line 257: Should the LOD be 1250-2500 instead of 1000-2500 copies/ml? Could the authors explain the calculations behind getting to these numbers?
- Line 263: Do the authors have a hypothesis for why Figure 3B displays several outliers when comparing individual to pooled samples? It would be good to include it in the discussion.
- Line 276-282: Could other pooling strategies be considered to improve detectability and streamline workflows depending on sample size?

5. Discussion:

- Line 377: Very minor – it would be good to rephrase saying “...reduced by only approximately...” as the placing of the word ‘only’ appears to change the connotation of the sentence. Further, for consistency and clarity, it would be good to express a percentage in line 377 as “nearly 100%” instead of “...nearly abrogate the total number...”
- Line 381: Very minor – consider rephrasing to something like “... lost without the timely turnaround of test results.”
- Line 460: The lack of a correlation for the N1 Ct value and the close contact spread is interesting and surprising. Any hypotheses for this observation? Is it possible the sample size is too small, or the nature of this study relatively subjective/nuanced?
- Line 477: In the discussion, while comparing fall and spring, it’s important to comment on the context by including regional and national trends in COVID-19 during the two semesters.

6. References: None

7. Figures and Tables:

- Figure 2B consider replacing ‘twenty’ with 20 for consistency.

Responses to Reviewer Comments

Other Modifications

- We updated lines 58-59 to include a more recent statistic about college vaccination rates as of September 2021.
- We included figure legends at the end of the manuscript (previously only in the PDF figures)

Reviewer 2 Comments:

- Lines 171-174 (Materials and Methods): Is it ok to run inter-run studies on different days? How does sample stability play a role and when is the sample gamma-inactivated if it's performed on a different day?
 - Yes, inter-run precision should be evaluated over multiple days to ensure stable assay and equipment performance across days, which is supported by guidelines from the Clinical Laboratory Standards Institute. We feel that most readers will understand the importance of verifying consistent assay performance over multiple days without further clarification of this point in the text, but we have added the following sentence in the Methods section to clarify how we ensured sample stability across multiple days: "For the latter, samples were aliquoted and frozen in individual tubes to be thawed on each test day to ensure uniform sample quality across days."
 - The samples used in the inter-run precision studies were not gamma-irradiated samples but rather were saliva samples from individuals with diagnostically confirmed SARS-CoV-2 infection. We have clarified this in the Methods section accordingly: "Intra-run and inter-run precision were assessed from replicates on a given plate or over multiple daily runs, respectively, using saliva from individuals with diagnostically confirmed SARS-CoV-2 infection."
- General comment about the Results: It would be good to mention upfront that there were 138 positives and what the split was between saliva and non-saliva tests to better orient the reader.
 - Yes, we have modified this wording as follows: "Collectively, we administered 43,884 saliva tests during the semester, flagging 114 (83%) of 138 total positives during the spring semester (Table 1)."
- Line 257 (Results): Should the LOD be 1250-2500 instead of 1000-2500 copies/ml? Could the authors explain the calculations behind getting to these numbers?

- 1000-2500 copies/mL is correct. The loss of signal (Ct 1.2 and 3.0) in pools of 5 and 10 correspond to LOD values achieved in Figure 2A of 1000 and 2500 copies/mL. We have modified this language to read: “Based on our LOD findings in Figure 2A, this would correspond to raising the input threshold to 1000-2500 copies/mL when these samples are diluted in pools of five and ten, respectively, compared to 500 copies/mL for samples run individually.”
- Line 263 (Results): Do the authors have a hypothesis for why Figure 3B displays several outliers when comparing individual to pooled samples? It would be good to include it in the discussion.
 - We suspect that this is because saliva is not entirely homogeneous. We have modified this in the results section to read: “Pooling generally resulted in higher N1 Ct values than when samples were run individually in the deconvolution run (Figure 3B), though several did not follow this trend suggesting that saliva may not contain homogenous viral sample”
- Line 276-282 (Results): Could other pooling strategies be considered to improve detectability and streamline workflows depending on sample size?
 - Certainly this is possible. Sample size, positivity rates, and turnaround time are all important considerations for pooling. This is addressed in the discussion, lines 409-422 of the marked-up manuscript.
- Line 377 (Discussion): Very minor – it would be good to rephrase saying “...reduced by only approximately...” as the placing of the word ‘only’ appears to change the connotation of the sentence. Further, for consistency and clarity, it would be good to express a percentage in line 377 as “nearly 100%” instead of “...nearly abrogate the total number...”
 - Thank you for helping us clarify this language. We have made these corrections.
- Line 381 (Discussion): Very minor – consider rephrasing to something like “... lost without the timely turnaround of test results.”
 - Good suggestion. This has been changed.
- Line 460 (Discussion): The lack of a correlation for the N1 Ct value and the close contact spread is interesting and surprising. Any hypotheses for this observation? Is it possible the sample size is too small, or the nature of this study relatively subjective/nuanced?
 - This result surprised us too, as our experimental number is large enough to at least identify trends, even if modest enough an effect so as not to be able to reach statistical significance. The finding that Ct’s were virtually

identical, and that regression was flat, indicates that to us that we were “catching” positives at various points along a continuum, and not necessarily at the very start of their infection. However, even when we conducted this analysis with twice a week testers (those testing more frequently), the results were the same.

- One possibility might from the challenges of congregate living of dorms and apartments were masks are often not worn among friends and roommates. In this case, viral load may be less important than overall time spent with an infectious individual. In other words, viral load may matter less with two hour interaction than someone who was a close contact for only 15 minutes.
 - Overall, we remain perplexed by this finding. Thus, we have not modified the discussion section on this point.
- Line 477 (Discussion): In the discussion, while comparing fall and spring, it's important to comment on the context by including regional and national trends in COVID-19 during the two semesters.
 - It is challenging to make apples to apples comparisons. National case numbers were higher in the spring semester, especially upon return, while local numbers were lower in the spring. On the other hand, we were much less strict with our regulations in the spring and activities like athletics were allowed to happen. While these are all important considerations, addressing this in the paper would require a lengthy and nuanced section that we feel would detract from the key implications of our report. Thus, we respectfully would like to decline this recommendation. We do however briefly mention the context in lines 258-261 of the Marked-up Manuscript. This content had previously been in the introduction.
 - Figure 2B consider replacing ‘twenty’ with 20 for consistency.
 - We have replaced “twenty” with “20”.

Reviewer 3 Comments:

- Major comment: The background is very thorough and almost reads as a complete study report. Therefore, I think to keep this as an introductory section, it could be work moving some key findings to the discussion, such lines 122 or 123-130 and lines 133-140. This would allow the authors to more clearly highlight the relevant background to their study and lead into what they did in response to this.
 - We thank the reviewer for this feedback and have abbreviated the contents of the original lines 122-130 to emphasize the success of the testing program while waiting until the Results and Discussion sections to discuss further outcomes.

- Figure 2 - why are the individuals only presumed negative? Were the samples tested prior to pooling to determine negative status?
 - We have removed the term “presumed” from the figure legend. The samples were shown to be negative for SARS-CoV-2 by our assay.
- Line 29 - suggest 'Limit of detection' rather than detection limit.
 - We have changed “detection limit” to “limit of detection”.
- Line 109 - 'FDA-authorized' (only has EUA, not full approval).
 - We have changed “FDA-approved” to “FDA-EUA (Emergency Use Authorization)” throughout the text.
- Line 185 - 3.75 ul of lysate is a deviation from the SalivaDirect protocol and could be mentioned here.
 - We have modified this to read: “3.75μL of heat-treated saliva was added for a final reaction volume of 15μL, which conserves reagents at three fourths the reaction volume used by Vogels et al.”
- Line 196 - 'by an FDA EUA (Emergency Use Authorization) test'
 - We have changed “by an FDA-approved EUA (Emergency Use Authorization) test” to “by an FDA-EUA test”.
- Line 227 - the authors could also consider stating 'the highest Ct values representing weakly positive samples' (or similar) for readers less familiar with RT-qPCR
 - We would like to respectfully decline making this modification. We have oriented the axis (reverse) to help those less familiar with RT-qPCR as well as clearly described in the text how to properly interpret the results.
- Figure 1 - suggest 'RT-qPCR' for consistency with the text
 - This has been updated.

September 21, 2021

Dr. Daniel R Sharda
Olivet Nazarene University
Biological Sciences
One University Ave
Bourbonnais, IL 60914

Re: Spectrum01089-21R1 (Routine, Cost-effective SARS-CoV-2 Surveillance Testing Using Pooled Saliva Limits Viral Spread on a Residential College Campus)

Dear Dr. Daniel R Sharda:

Your manuscript has been accepted, and I am forwarding it to the ASM Journals Department for publication. You will be notified when your proofs are ready to be viewed.

Sincerely,

Tulip Jhaveri
Editor, Microbiology Spectrum
